

# Runoff reaction from extreme rainfall events on natural hillslopes: A data set from 132 large scale sprinkling experiments in south-west Germany

Fabian Ries[1], Lara Kirn[2], Markus Weiler[1]

[1]Chair of Hydrology, University of Freiburg, 79098 Freiburg, Germany
[2]Unger Ingenieure, 79098 Freiburg, Germany

*Correspondence to*: Markus Weiler (markus.weiler@hydrology.uni-freiburg.de)

**Abstract.** Pluvial or flash floods generated by heavy precipitation events cause high economic damages and loss of life worldwide. As discharge observations from such extreme occurrences are rare especially on the scale of small catchments or

even hillslopes, data from artificial sprinkling experiments offer valuable information on runoff generation processes, overland and subsurface flow rates and response times. We conducted 132 large-scale sprinkling experiments on natural hillslopes at 23 sites with different soil types and geology on pastures and arable lands within the federal state of Baden-Württemberg in south-west Germany. The experiments were realized between 2016 and 2017. Simulated rainfall events of varying durations were based on a) the site-specific 100-year return periods of rainfall with different durations and b) the

maximum rainfall intensity observed locally. The 100 m² experimental area was divided into three individual plots and overland and subsurface flow, soil moisture and water level dynamics in the temporarily saturated soil zone were measured at 1-minute resolution. Furthermore, soil characteristics were described in detail for each site. The data was carefully processed and corrected for measurement errors and combined to a consistent and easy to use database. The experiments revealed a large variability of possible runoff responses to similar rainfall characteristics. In general, agricultural fields

produced more overland flow than grassland. The latter generated hardly any runoff during the first simulated 100-year event on initially dry soils. The dataset provides valuable information on runoff generation variability from natural hillslopes and may be used for the development and evaluation of hydrological models, especially those considering physical processes governing runoff generation during extreme precipitation events. The dataset presented in this paper is freely available from the FreiDok plus data repository (https://doi.org/10.6094/UNIFR/149650, Ries et al., 2019).

## 1 Introduction

Pluvial floods (sometimes in an extended context also referred to as surface water floods or flash floods) originate from extreme, often small-scale convective rainfall events that can exceed the infiltration capacity and consequently lead to ponding and overland flow (Bernet et al., 2017). Such events can cause tremendous economic damage. Pluvial floods in urban areas receive significant public attention, possibly because of an elevated probability of a high number of affected





people and large economic damages caused by a single event, even though rural areas are equally affected. Climate change is expected to increase the frequency and intensity of extreme precipitation events (IPCC, 2013). However, according to a study from the UK, population growth, surface sealing and urbanization may contribute even more to an increased flood risks from extreme rainfall events (Houston et al., 2011). In recent years, exceptionally devastating pluvial floods in

Germany and entire Europe (see Bernet et al., 2017 for some examples) have intensified the awareness of the risk associated with such events and put pressure on water management and communal decision makers to better predict possible flood events and identify flood-risk prone locations distant from permanent watercourses (Rosenzweig et al., 2018). It has also resulted in the development of adaption strategies to increase resilience in urban areas (e.g. Carter, 2011; Haghighatafshar et al., 2014, Jiang et al., 2018) and in rural areas to a minor extent. However, handling pluvial flood risks will remain a

challenging task for several reasons:

1) Heavy convective rainfall events are difficult to forecast with adequate lead time, making traditional warning and mobile flood protection systems inadequate

2) The spatial and temporal distribution of pluvial floods: they can occur far from permanent watercourses and with great infrequency. Both reduces public awareness and perceived personal risk

3) Finally, because continuous growth of infrastructure increases the area potentially affected from pluvial floods

So far especially in rural areas, hazard response plans are more common than effective pre-disaster mitigation strategies (Frazier et al., 2013). The latter requires information on flood risk in rural communities, especially concerning pluvial floods not directly connected to watercourses, which is rarely available.

Over the past decades, research on runoff generation has led to a considerable expansion of knowledge on the diversity of

runoff generation processes and influencing factors at the hillslope and catchment scale (Beven, 2004). While many processes are well understood in theory, they are rarely considered in hydrological models or even operational flood forecasting. The question which process may be dominant under which conditions and their dependency on surface and subsurface properties is still one challenging aspect. Another is the fact that relevant processes are changing in dependency on spatial and temporal scales and require certain input data at the right resolution. Classical extreme value statistics to

predict return periods of floods are based on long streamflow time-series at discharge gauges either by probabilistic approaches or hydrological models calibrated with streamflow time-series. Those gauges are typically installed at larger river basins, where small-scale convective rainfall events of high intensity and short duration are not the main driver of floods.

For small catchments, one approach is the estimation of dominant runoff generation processes according to decision schemes based on surface and soil characteristics (e.g. Scherrer and Naef, 2003; Markart et al., 2006). Another is the development of

hydrological models that actually represent important processes affecting infiltration and runoff generation during extreme precipitation events on an appropriate spatial and temporal extent (e.g. Steinbrich et al., 2016). Model development as well



as model evaluation would benefit from long-term runoff time-series from small catchments, hillslopes or - in case such data is not available - from experiments simulating extreme rainfall events.

To address this gap, we contribute an extensive database from numerous field experiments on runoff generation during simulated extreme precipitation events. The data was already used to validate a pluvial flood model in the federal state of
Baden-Württemberg, Germany. We encourage the further use of the data for runoff generation research and the development and evaluation of process-based hydrological models advancing the important topic on risk assessment from pluvial flooding.

## 2 Experimental sites

Experimental sites were selected with the aim of covering a large variety of soil types present within the federal state of
Baden-Württemberg (Germany) on pastures and arable land. Beside sealed urban areas, both land use types are known to respond with high runoff coefficients to intense precipitation events. For technical reasons, the selected experimental sites were restricted to locations with a minimum slope of 5 % to ensure free drainage of generated overland flow. In addition, the study site had to be in close proximity to build-up areas or a drinking water supply line to guarantee the required water amounts of approximately 130 m$^3$. Between August 2016 and October 2017, 132 sprinkling experiments were realized at 23
locations - 13 on pastures and 10 on arable sites. Location, land use, geology and soil characteristics of the selected experimental sites are summarized in Tab. 1. To directly compare the effect of the two land uses on runoff response, 12 of the 23 locations are paired sites (gray background in Tab. 1) with different land use but presumably comparable soil characteristics, geology and development due to their immediate proximity (less than 100 m). Figure 1 maps the location of the experimental sites and the distribution of soil types within the federal state of Baden-Württemberg according to the soil
map 1:50 000 (BK50) (LGRB, 2017). According to the BK50, some soil types rarely occur (less than 0.1 % of the area; shaded area in the soil triangle in Fig. 1) within the federal state of Baden-Württemberg. Grain size distribution and soil types determined in the laboratory from samples taken at the experimental sites may differ from those documented in the BK50 soil map.





**Table 1:** Site characteristics of the experimental locations.

| No | Location name | Land use | Height [m asl] | Slope [%] | Geology [LGRB, 1998] | Sand[a] [%] | Silt[a] [%] | Clay[a] [%] |
|---|---|---|---|---|---|---|---|---|
| 1 | Schönberg | pasture | 371 | 12 | Middle Keuper | 15 | 44 | 42 |
| 2 | Wildtal | pasture | 278 | 18 | Gneiss | 5 | 67 | 28 |
| 3 | Freiburg | pasture | 303 | 16 | Middle Red Sandstone | 36 | 45 | 20 |
| 4 | Freiburg | arable land | 299 | 16 | Middle Red Sandstone | 32 | 35 | 33 |
| 5 | Freiamt | arable land | 431 | 14 | Lower Shell Limestone | 7 | 69 | 24 |
| 6 | Freiamt | pasture | 430 | 21 | Lower Shell Limestone | 5 | 56 | 39 |
| 7 | Opfingen | arable land | 228 | 14 | Loess and Loam | 13 | 64 | 23 |
| 8 | Seelbach | arable land | 245 | 16 | Middle Red Sandstone | 52 | 29 | 19 |
| 9 | Seelbach | pasture | 249 | 21 | Middle Red Sandstone | 49 | 36 | 15 |
| 10 | Sankt Märgen | pasture | 850 | 32 | Diatexite | 51 | 14 | 35 |
| 11 | Wehingen | pasture | 795 | 18 | Brown Jurassic | 19 | 40 | 41 |
| 12 | Gosheim | pasture | 868 | 19 | Brown Jurassic | 5 | 45 | 50 |
| 13 | Gosheim | arable land | 847 | 11 | Brown Jurassic | 4 | 59 | 38 |
| 14 | Bonndorf | pasture | 821 | 27 | Lower Shell Limestone | 7 | 49 | 44 |
| 15 | Zimmern | arable land | 670 | 14 | Jurassic | 25 | 43 | 33 |
| 16 | Zimmern | pasture | 694 | 12 | Jurassic | 31 | 43 | 27 |
| 17 | Aasen | pasture | 714 | 14 | Cley Keuper | 25 | 48 | 27 |
| 18 | Aasen | arable land | 715 | 12 | Cley Keuper | 16 | 39 | 45 |
| 19 | Baiersbronn | pasture | 596 | 21 | Middle Red Sandstone | 60 | 20 | 20 |
| 20 | Raithaslach | arable land | 590 | 9 | Wuerm moraine sediment | 52 | 33 | 15 |
| 21 | Neckartenzlingen | arable land | 339 | 14 | Sandstone Keuper | 10 | 48 | 42 |
| 22 | Waldstetten | pasture | 406 | 12 | Opalinus Clay | 8 | 48 | 45 |
| 23 | Haidgau | arable land | 682 | 14 | Riss moraine sediment | 38 | 36 | 26 |

[a] Grain size distribution as an average from soil samples in 10, 30 and 50 cm depth. Soil texture classes follow the German particle size classification with sand (2000 - 63 μm), silt (63 - 2 μm) and clay (< 2 μm). Soil texture analysis is described in Sect. 3.4. Paired sites with comparable soil characteristics but different land use are shaded grey.

**Figure 1:** Location of experimental sites and distribution of soil types in the federal state of Baden-Württemberg according to the soil map BK50 (LGRB, 2017). Site textures in the soil texture triangle are shown according to particle size measurements in the laboratory.



## 3 Field experiments

Prior to conducting the sprinkling experiments, we developed a mobile rainfall simulator. Its dimensions were selected to cover a large representative area of a hillslope and at the same time permit the use of the public water supply network, which is often restricted in terms of maximum flow rate and water pressure. The rainfall simulator was further designed to allow for an even rainfall distribution on a wide range of rainfall intensities. An illustration of the entire experimental setup is displayed in Fig. 2.

### 3.1 Water supply and rainfall simulation

At all experimental sites, water was taken from the public drinking water network by connecting 3-inch firehoses to close-by hydrants. The mobile rainfall simulator consisted of 12 sprinklers with a circular footprint (Senninger, XCEL Wobbler UP3 Top), which are characterized by a uniform spatial distribution and a near natural drop size (van Meerveld et al., 2014). The upward-sprinklers were attached to aluminum rods at a height of 1.8 m and arranged along two rectangles (Fig. 2). To reduce boundary effects, the irrigated field (15x15 m) covered an area more than twice the size of the actual experimental runoff plot. A pressure regulator (10 psi) attached to each sprinkler guaranteed a constant pressure and flow rate at each sprinkler independent of the location. By attaching different nozzles, a wide range of sprinkling intensities could be simulated, reaching up to 170 mm/h. The simulation of high rainfall intensities required flow rates up to 500 l/min – a circumstance that drastically reduced the potential experimental sites to locations close to main supply lines with sufficiently high water pressure. Rainfall distribution in space and time was recorded with 6 automatic precipitation gauges (RG1-RG6) (Onset, HOBO RG-3) and 11 rainfall totalizers (C1-C11), which were read and emptied manually after each experiment.

### 3.2 Plot setup, instrumentation and runoff measurements

The experimental plot of 10x10 m was divided into three subplots of equal size (A, B, C) each with a width of 3.33 m and a length in slope direction of 10 m. To confine the runoff contributing area, thick plastic sheets of 15 cm height were inserted approximately 5 cm into the soil at the plot's upper margin and sides, as well as in between the subplots defining the plot boundaries. During the experiments at the first 5 experimental sites, accumulation of water was sometimes observed in depressions above the upper plot boundary. To avoid possible inflow from this area, and to keep the plot water balance as closed as possible, a plastic cover (5x10 m) was placed above the experimental plot from this point on. To measure overland flow (OA), a trench was excavated on the lower end of the plot and a sturdy plastic sheet was inserted into the vertical profile wall which diverted (near) overland runoff (max 5 cm depth) into rainfall gutters and from there, via closed pipes and separated for each subplot, to the actual runoff measurement device (Fig. 3). The trench was covered with plastic sheets to avoid direct rainfall input into the runoff gutters. At plot B, the trench was excavated to a depth of 40 cm and subsurface flow (SSF) was conveyed with a drainage mat and measured separately.





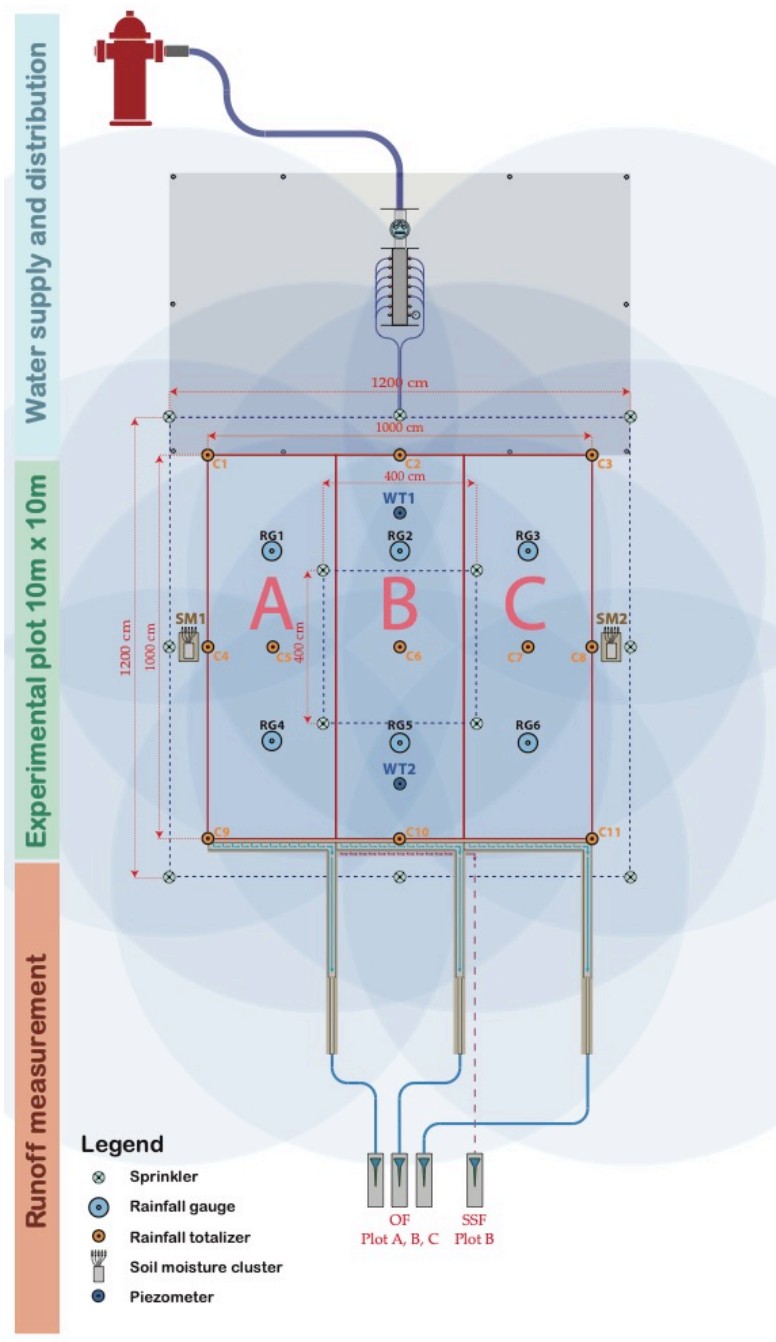

**Figure 2:** Illustration of the experimental setup with water supply and distribution, experimental plot, and devices for the quantification of overland flow and subsurface runoff. Runoff was measured from an area of 10 x 10 m while the irrigated area covered approximately 15 x 15 m to reduce boundary effects. The blue shaded circles illustrate the approximate extent of each sprinkler.

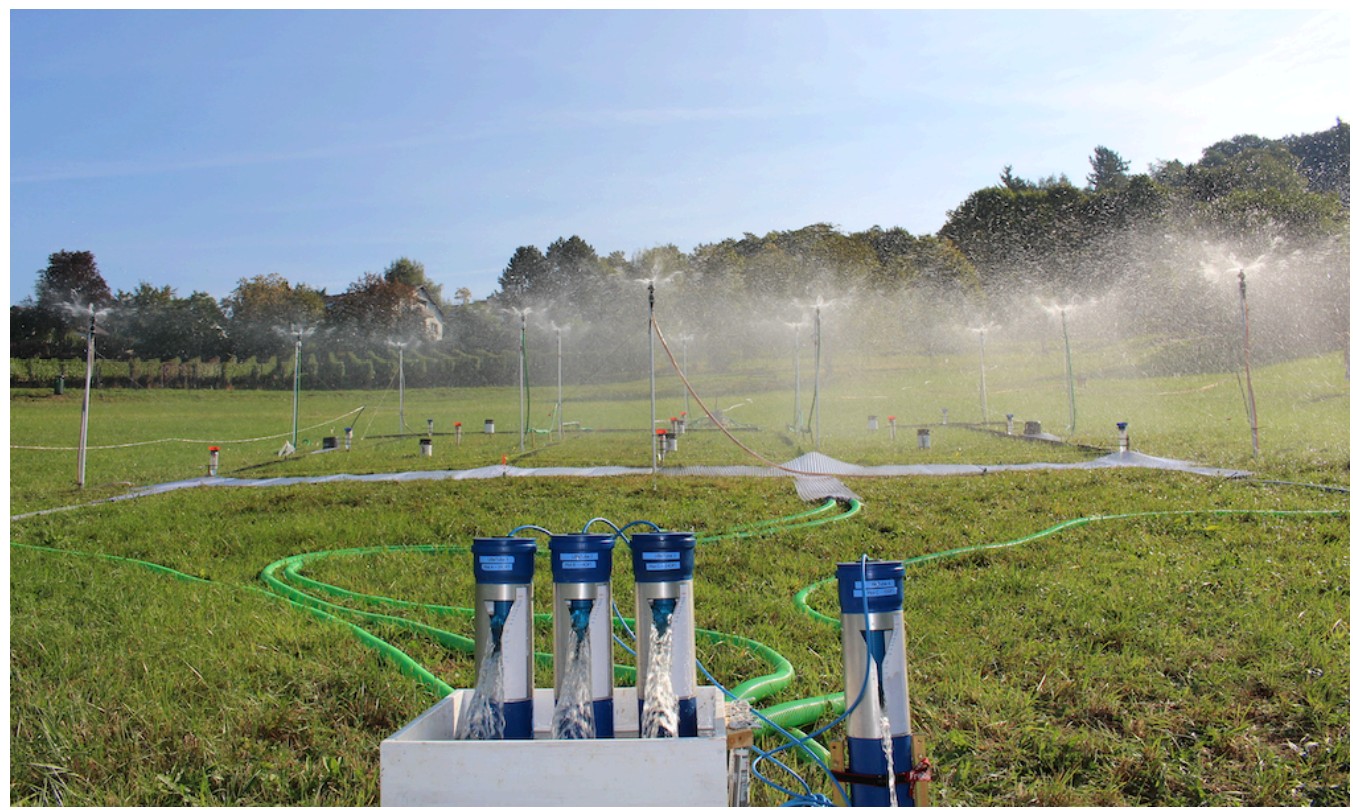

**Figure 3:** Exemplary setup of the field experiments. The foreground shows the measurement device for surface and subsurface runoff and the connecting lines (green hoses) from the trench covered with plastic sheets. Visible in the background is the actual experimental plot divided into the three subplots with rainfall gauges and the sprinkling system.

Overland flow from the three plots and subsurface runoff was conducted to the Upwelling Bernoulli Tube (UBeTube), a measurement device adopted from Steward et al. (2015) where water enters laterally into a plastic tube (one end closed) and raises until it leaves through a double trapezoid-shaped stainless-steel weir. The device's maximum flow rate capacity is 120 l/min or 215 mm/h on a surface area of 33.3 m². The water level in the tubes was recorded with an air pressure compensated piezometer (HT – Hydrotechnik, 575-II) with an accuracy of 1 mm. The stage-discharge relationship from Steward et al. (2015) was slightly modified with a correction factor determined through calibration experiments. Large fluctuations of the water level inside the tube caused by trapped air in the conducting line or temporal clogging of the lower trapezoid by washed in sediments and debris were manually identified from the runoff time series and corrected. Two soil moisture profile clusters with sensors in 5, 10, 20, 30 and 40 cm depth (Trübner, SMT 100) were installed on both sides of the experimental plot (SM1, SM2) within the sprinkling area. To measure the depth of a possibly developing perched water table, wells (Eijkelkamp, Micro-Diver) were installed in two-inch filter tubes inserted between 0.5 and 1 m into the soil within the experimental plot, 1.5 m above and below the lower respectively upper plot boundary (WT1, WT2). The tubes



were sealed with duct tape from the bottom and with clay material at the top of the soil surface to reduce possible water inflow from overland flow. Wind speed, temperature, humidity and solar radiation were measured close to the experimental site during the entire period. Attention was paid during the installation of all field equipment to reduce the disturbance of the experimental plot.

## 3.3 Experimental procedure

All 132 experiments at the 23 different sites were conducted between August and November 2016, and May and October 2017 on days with none or only minimal amounts of natural precipitation. The installation and realization of the experiments at each site took about four days and followed essentially the same pattern:

Day 1: Set-up the experimental site

Day 2: Completion of set-up and experiment 1 (60 min, 100 y return period)

Day 3: Experiment 2-4 (60, 30, 15 min duration, 100 y return period) and experiment 5 (180 min duration, extreme scenario)

Day 4: Experiment 6 (60 min, extreme scenario) and disassembly of experiment set-up.

The intensity of the simulated precipitation events for each site were selected according to a statistical analysis of observed station data in the federal state of Baden-Württemberg (LUBW, 2016). The experiments 1–4 (1 h, 1 h, 30 min, 15 min) represent a location specific rainfall event with a return period of 100 years and the experiments 5 and 6 (180 min, 60 min) correspond to a "worst-case scenario". For this scenario, we applied 138 mm in 3 h and 106 mm in 1 h corresponding to the highest ever in Baden-Württemberg observed rainfall event for the respective duration. Commonly, there were 12 hours without simulated rainfall between experiments 1 and 2 and experiment 5 and 6. Experiments 2–5 usually took place at the same day with a break of at least 15 min between the single trials. Overland flow stopped usually within this period while the recession of the subsurface runoff often extended into the start of the follow-up experiment. At a few locations we had to interrupt the usual sequence of experiments for up to 2 days due to strong winds that would have altered the distribution of the simulated rainfall.

## 3.4 Field soil description and laboratory analysis

At each location, the surface characteristics and soil-hydrological properties (e.g. bulk density, root density, stone content and the number of macropores with a size larger than 2 mm and 5 mm) were recorded for the depths of 10 and 30 cm according to DWA (2018). Soil samples in 10, 30 and 50 cm depth were taken and analyzed for grain size distribution in the laboratory. Samples were dried at 105 °C, crushed and sieved to exclude particles larger than 2 mm. In a next step the organic matter was removed with hydrogen peroxide, dispersed and the silt fraction (2–63 μm) determined with a particle size analyzer (Pario, Meter-Environment) which works on the basis of gravitational settling. Finally, the sand fraction (63–2000 μm) was sieved out and the clay-content (< 2 μm) was calculated as a remainder.



## 4 Data quality, processing and description

A total of 132 out of 138 intended experiments were executed as planned. The remaining 6 experiments (experiment 3 at location 4, experiments 4-6 at location 8 and experiment 5 and 6 at site 23) had to be stopped due to extreme erosion at the runoff trench or high sediment input to the UBeTubes and associated measurement errors. All data measured in the field were checked for inconsistencies and compiled into a single data set. The following provides a brief overview of the collected data and summarizes basic results.

### 4.1 Rainfall

Systematic measurement errors of the precipitation gauges caused by the high intensities were corrected with a dynamic correction factor determined in the laboratory under controlled conditions. Precipitation data from the tipping bucket rain gauges was discretized using the temporal distribution recorded at the rain gauges. The spatial distribution was then determined with the inverse distance method within the R package phylin (Tarroso et al., 2019). For each minute, spatial mean values were calculated for the three sub-plots as well as the entire experimental plot. Table 2 shows the range of target and actually simulated intensities for the individual experiments. The simulated precipitation intensity accounted for 85 % to 126 % of the target rainfall intensity with an average of 102 %. Deviation caused by the deformation of the sprinkling area and uneven distribution due to wind effects, variations in water pressure from the supply line and the step-wise adjustment of the flow rate with limited amount of nozzles sizes.

Table 2: Experiment characteristics, target and simulated intensities for all experimental sites.

| Experiment No | Duration and return period | Target intensities (mm/h) | Simulated intensities (mm/h) |
|---|---|---|---|
| 1+2 | 60 min; 100 y | 41-69 | 42-76 |
| 3 | 30 min; 100 y | 80-115 | 82-130 |
| 4 | 15 min; 100 y | 108-173 | 110-172 |
| 5 | 180 min; "worst case" | 46 | 44-53 |
| 6 | 60 min; "worst case" | 106 | 99-126 |

The rainfall uniformity coefficient of the individual experiments calculated according to Christiansen (Eq. 1) ranged from 75% to 93% with an average of 87%, which can be considered a good uniformity (Merriam and Keller, 1978).

$$CU = 100 \cdot \left( 1 - \frac{\sum |x - \bar{x}|}{(\bar{x} \cdot n)} \right) \tag{1}$$

CU  = Christiansen Coefficient (%)

x     = Measured precipitation (mm)
x̄     = Mean measured precipitation (mm)
n     = Number of observations

Deviation from the mean precipitation applied in all 132 individual experiments ranged between -17 % and +24 % with a

concentration in the middle of the plot and a reduction towards the edges (Fig. 4).

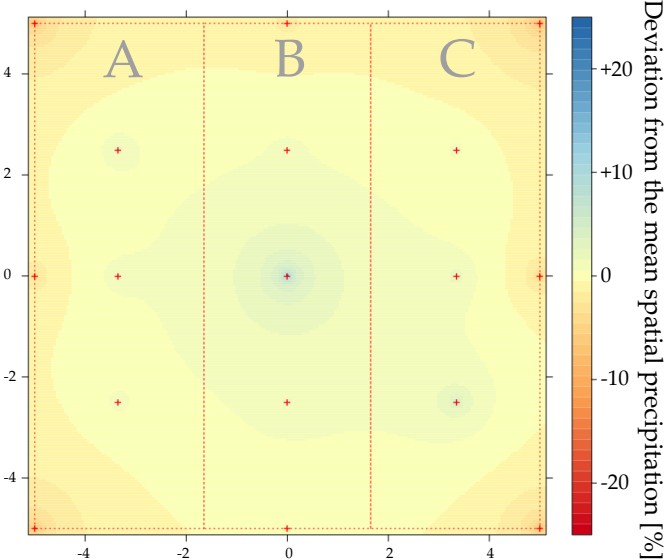

**Figure 4:** Interpolated rainfall distribution from all 132 experiments as a deviation from the spatial mean in percent.

**4.2 Runoff measurements**

An example of the measured variables provided in the data set at two different experimental sites and for all 6 individual experiments (E1-E6) on grassland (site 11) and agricultural field (site 20) are shown in Fig. 5 and Fig. 6. Figure 5 shows the delayed runoff response on the grassland site where overland flow was first observed 30 minutes into the second 100-year rainfall event. The second example (Fig. 6) in contrast, shows overland flow starting only 10 minutes following the initiation

of the first sprinkling experiment and quickly reaching a rate close to the rainfall input.





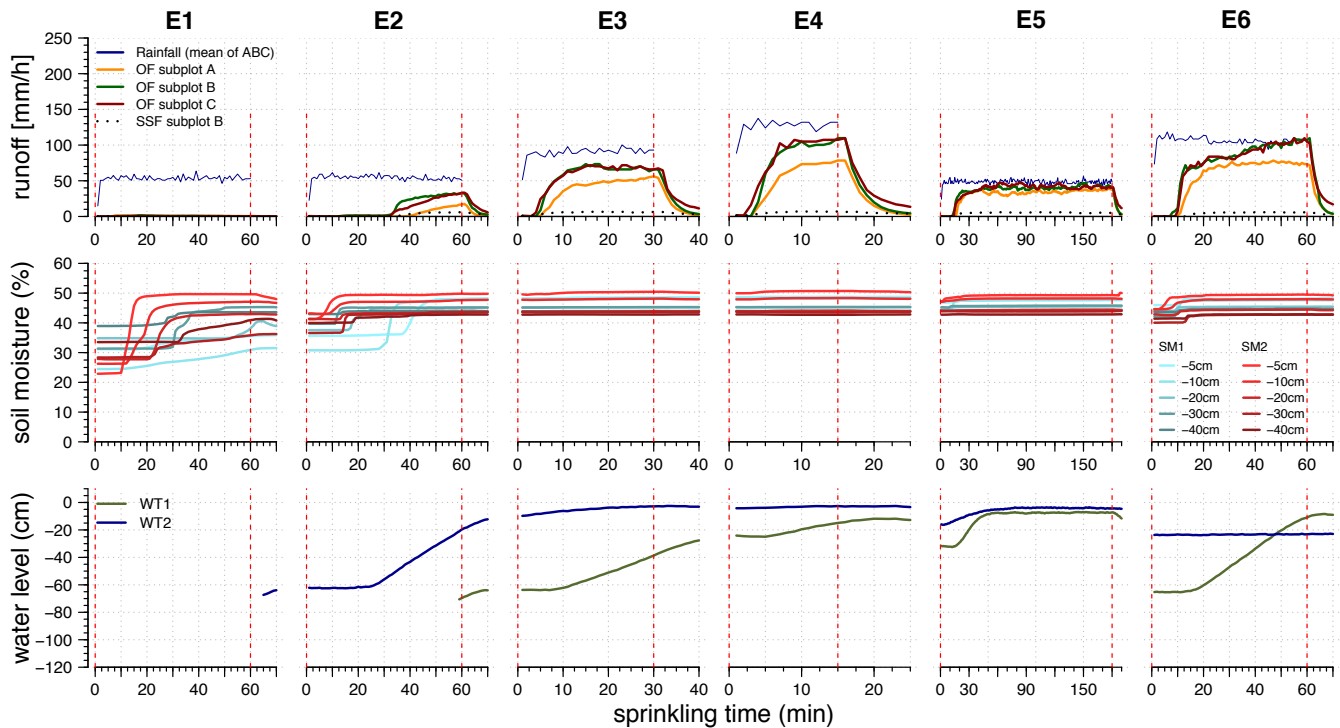

**Figure 5:** Example for runoff reactions, soil moisture and temporary saturated soil zone for experimental site 11 on pasture for the six single simulations of extreme rainfall.

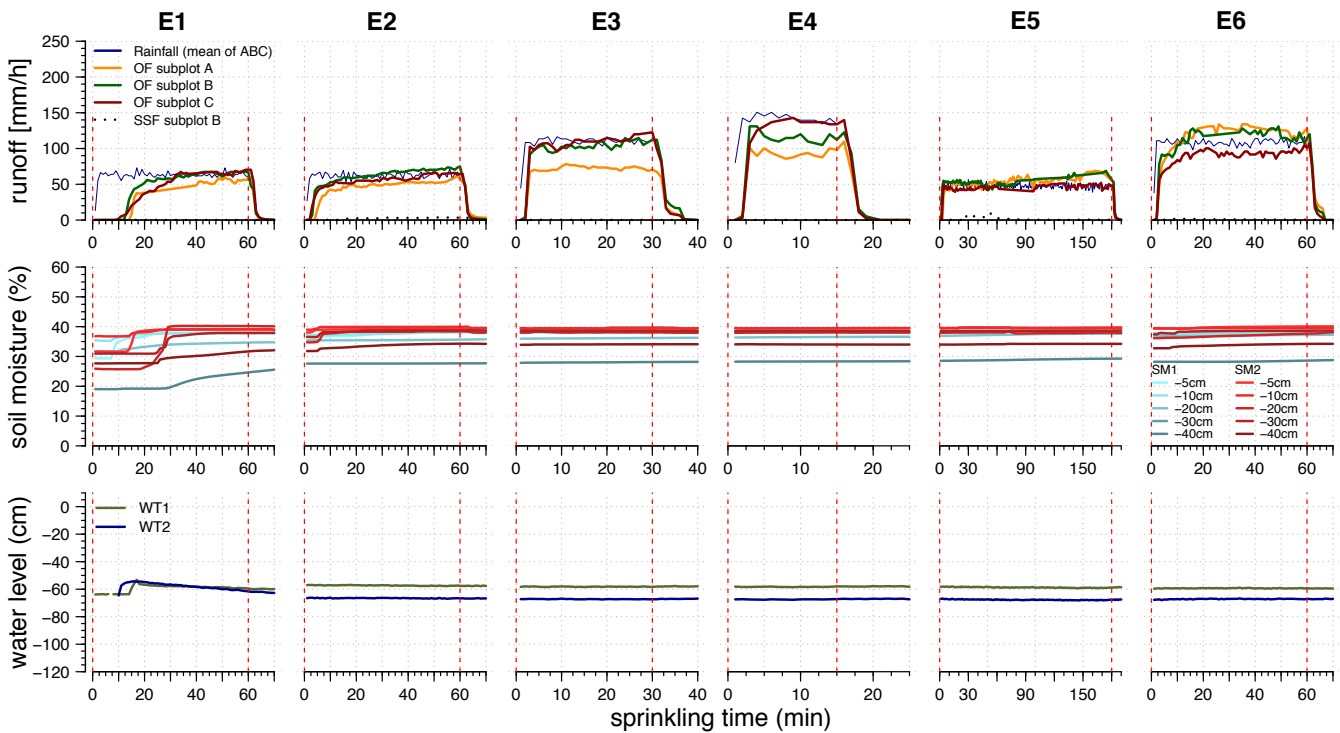

**Figure 6:** Example for runoff reactions, soil moisture and temporary saturated soil zone for experimental site 20 on a recently harvested corn field for the six individual simulations of extreme rainfall. The runoff hydrograph at subplot A in Experiment 3 and subplot A and B in experiment 5 and 6 illustrate measurement errors mentioned in chapter 4.2.

## 4.3 Runoff variability

The observed overall runoff coefficients from all experiments at individual sites ranged between 1 % and 87 % of the applied precipitation and between 0 % and 100 % for individual experiments compared across all sites, respectively. Figure 7 shows the large variability of runoff reactions and the difference between the two land use types - grassland and field. The comparison of runoff coefficients and hydrographs between experiments 1 and 2 with the same intensity and duration shows the effect of soil moisture on runoff rates for the simulated extreme rainfall events (Fig. 8).

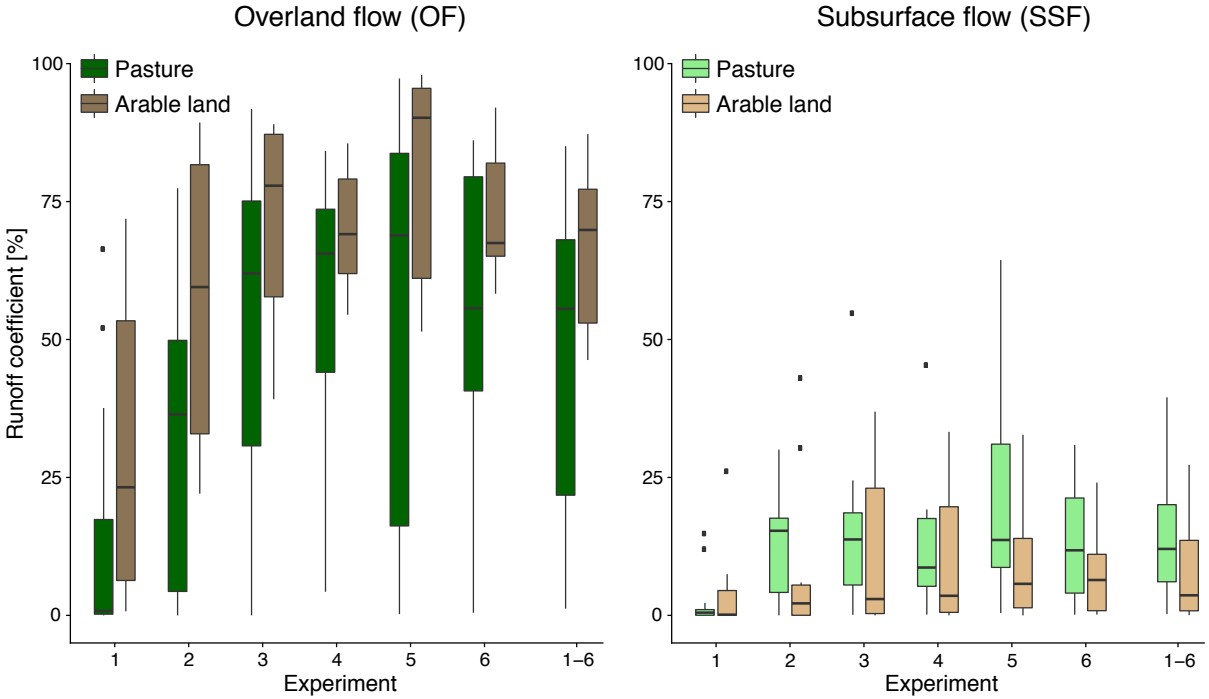

**Figure 7:** Overland and subsurface runoff coefficients from all experimental sites separated by experiment number (1 to 6) and for all experiments (1-6) and the two land use types pasture and arable land.

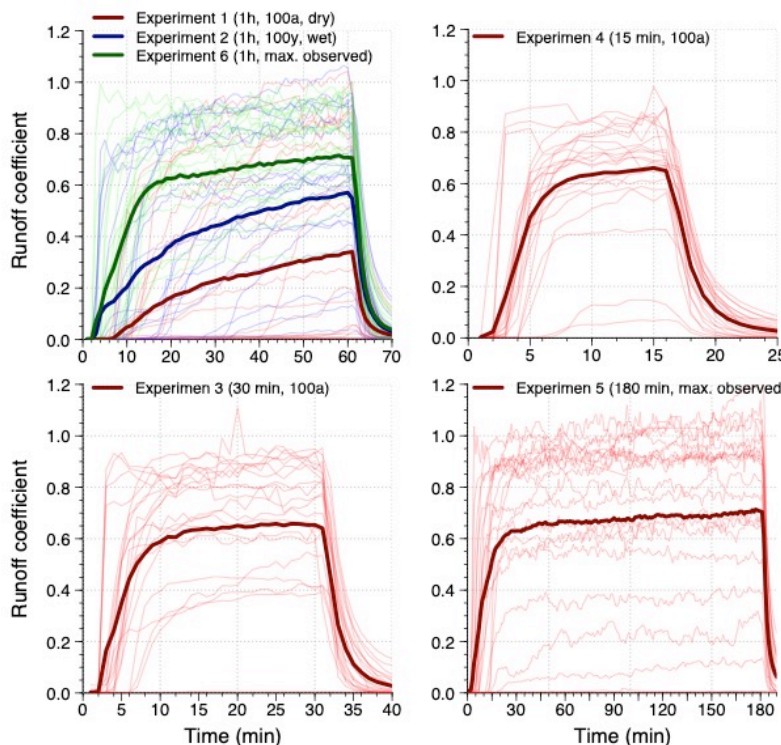

**Figure 8:** Overview of hydrographs from all experiments separately displayed for the individual durations and return intervals.



## 5 Data availability and structure

The data set described in this paper is publicly available from the FreiDok plus data repository (https://doi.org/10.6094/UNIFR/149650, Ries et al., 2019). The structure and contents of the single files are summarized in Tab. 3. All observed variables (e.g. precipitation, soil moisture and runoff) of the individual experiments are combined into one time series file containing information on location and experiment number providing the possibility of a simple filtering by multiple site characteristics e.g. with the filter option of the dplyr-package in R (Wickham et al., 2018) or similar approaches. The time series of each location starts with the day of the first experiment of the respective location.

Table 3: Overview of the single files provided in the FreiDok data repository. All files are packed in a single zip-File.

| File name | Content |
|---|---|
| 0_README.txt | Information on the structure and the content of the data set |
| 1_site_data.txt | Data on experimental site characteristics (e.g. topography, land use, vegetation cover and soil characteristics) |
| 2_event_data.txt | Data on experiment characteristics and results from the individual simulations (e.g. duration, return period, intensity, start and ending time, cumulative rainfall and runoff amounts) |
| 3_time_series_data.txt | Combined time series of the observed variables of all experiments in a resolution of 1 minute containing data on simulated rainfall intensity, overland flow, subsurface runoff, observed soil moisture, depth of temporally saturated conditions and meteorological parameters |
| 4_soil_images | Folder containing images of the surface and horizontal soil profiles in 10 and 30 cm depth of the individual experimental sites |

**Author contributions**

The design of the field experiments was elaborated by all authors. Fabian Ries and Lara Kirn realized the experiments, and Fabian Ries prepared the data and the manuscript with contributions of all co-authors.

**Competing interests**

The authors declare that they have no conflict of interests.

**Acknowledgements**

This study was funded by the State Office for the Environment Baden-Württemberg (LUBW). We would like to thank the local water supplier, farmers and landowners for the constructive cooperation and the permission to realize the field



experiments on their lands. We owe gratitude to Frank Bartmann from BNNETZE for advice on water abstraction from the public drinking water networks and to Dr. Frank Waldmann from the Federal State Office of Geology, Raw Materials and Mining (LGRB) for discussions on soils and soil-hydrological characteristics in the study area. Especially we would like to thank Marvin Lorff, Haytham Zireeni, Johanna Geilen, Laura Vecera, Annemarie Hoffmann, Lisa Kiemle, Carolin Winter,

Jakub Jeřábek, Maja Gensow, Birgit Müller, Emil Blattmann, Jonas Zimmermann and Britta Kattenstroth for greatly supporting the laborious field campaigns.

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
