# Peer review of "Runoff reaction from extreme rainfall events on natural hillslopes: A data set from 132 large scale sprinkling experiments in south-west Germany"

_Earth System Science Data, 2019_

## Referee Comment (RC1) · Anonymous Referee #1 · 4 Sep 2019

General comments:

The authors present a very valuable dataset on sprinkler experiments in the Alps. The comprehensive dataset (freely available at FreiDok plus data repository, https://doi.org/10.6094/UNIFR/149650) is provided in one single zip file, file format is txt with all necessary Metadata and easily importable to R, MATLAB, MS Excel (that is what I have tested ïĄŁ). The provided data are of high value for experimentalists, modelers (rainfall-runoff) but also soil- and ecohydrologists. Provided pictures of soils and soil cover (i.e. vegetation) are very valuable and can help the user of the datasets

to at least estimate partly missing information. Nevertheless, my main suggestion is to try hard to add further data on topography (slope angle, slope aspect, elevation) and plant traits (LAI, vegetation height, biomass dry weight, . . .). Following few comments, which might further improve this study and should be addressed in the revised version of the manuscript:

* I like the test of rainfall distribution and the fact that the study addresses real rainfall provided at several points in field plots by using precipitation gauges. These information are very valuable to trust the data because of knowledge on drop eenergy. I would suggest to add a table in the manuscript following "site-data.txt" so that people quickly can check if the data is suitable for the research questions of the respective reader, especially from the biological characteristics point of view.

* Pastures are grassland, so at least provide estimation of vegetation height at the timing of rainfall experiments (because of interception and influence on rainfall distribution on the ground). It would be helpful to have some key species mentioned or perfect if you have a vegetation survey of the plots (not mandatory, but any information on that might be very helpful for the user of your data).

* Please discuss in a broader context which soil parameters (i.e. soil organic matter SOM, soil stone content) or plant parameters might be important and/or influence infiltration behavior as well as water storage capacity. Also address the kinetic energy of your water compared to natural rainfall and how this might impact your results, infiltration, and possible hydrophobic effects - this helps the reader to better undertsand the complex issue of sprinkler experiments.

To cover abovementioned issues I recommend to have a look into papers from Abdallah Alaoui (CH) or Christian Newesely (AUT) or Georg Leitinger (AUT) and many others. Not necessarily cite them but to gain a broader view on the addressed issue.

Specific comment:

* Figure 1: what do the hatched areas mean? (I assume to highlight which soil types have been investigated / covered, but then: why is there a red dot in a hatched area?). Please clarify!

Finally my apologies for providing this review at the last day of the deadline.

---

## Referee Comment (RC2) · Anonymous Referee #2 · 23 Sep 2019

General comments

The manuscript presents the results of extensive hillslope scale sprinkling experiments on 23 test sites in south-west Germany. It represents a valuable contribution to hillslope hydrology and to hydrology of pluvial floods. The data set provides information on the test sites as well as measurements of sprinkling input and resulting overland flow and subsurface flow and soil moisture response.

The presented experimental data set is clearly of interest for the scientific community.

[Figure]

The measurements were carried out in a intelligent way, the data set is in most parts ready to use and the manuscript is in most parts well-written and the figures show high quality.

The manuscript presents the experiments in a concise way. It would improve from more details on the test sites and from a more detailed description of the results. The manuscript lacks completely any interpretation and comparisons to similar experiments.

Specific comments

The authors should mention that similar sprinkling experiments on the hillslope scale were already carried out by others. Please name some of the most important sprinkling experiment studies. How does your experimental set-up and your results compare to the findings of others?

It would be useful to have more information on the test sites, e.g. what was the soil depth above the soil-bedrock-interface? Was the bulk density and the density of macropores evaluated?

The manuscript would improve from a more detailed description of the results. In particular, a more detailed overview and a comparison of the results on the different test sites would be helpful.

In addition some interpretation of the results would be very interesting. In the introduction section pluvial floods are mentioned as a motivation for the study. What is the interpretation of your results with regard to pluvial floods? The runoff coefficients show large differences at the different sites. How can this be explained? The runoff coefficients show partially extremely high values of 100% (and more?). How can this be explained?

Technical corrections

Equation 1 and axis labels of Figure 4 are not readable.

The data set is partially incomplete with regard to soil moisture data, precipitation input and information on site and experiment number.

---

## Author Response (AR1)

**Reply to Review #1:**

Dear reviewer,

Thank you for your helpful comments and recommendations to our manuscript. Please find your questions and comments marked as e.g. "R1.C1" followed by our reply marked as e.g. "R1.A1" including a description of changes to the manuscript.

Best regards
Fabian Ries (on behalf of all co-authors)

**R1.C1:** My main suggestion is to try hard to add further data on topography (slope angle, slope aspect, elevation) and plant traits (LAI, vegetation height, biomass dry weight, …).

> **R1.A1:** Some of the requested data on topography ("Elevation" and "Slope") and vegetation ("Plant_coverage") are provided in the data file "1_site_data.txt". We will add available information on slope aspect and vegetation height to the data file "1_site_data.txt". LAI and biomass dry weight were not measured explicitly in the field, therefore we are unfortunately unable to provide this particular information.

**R1.C2:** I like the test of rainfall distribution and the fact that the study addresses real rainfall provided at several points in field plots by using precipitation gauges. These information are very valuable to trust the data because of knowledge on drop energy. I would suggest to add a table in the manuscript following "site-data.txt" so that people quickly can check if the data is suitable for the research questions of the respective reader, especially from the biological characteristics point of view.

> **R1.A2:** We did not measure the kinetic energy and size of the simulated raindrops but we refer to another study using a very similar sprinkling system (see chapter 3.1 "Water supply and rainfall simulations") that confirm near natural rainfall conditions of our sprinkling devices. Observations of the all rainfall gauges and rainfall collectors are included as cumulative values for the individual experiments in table "2_event_data.txt" and in minute time interval in "3_experiment_time_series.txt". Concerning further information on general characteristics of the individual experimental sites please see our reply to R1.A1.

**R1.C3:** Pastures are grassland, so at least provide estimation of vegetation height at the timing of rainfall experiments (because of interception and influence on rainfall distribution on the ground). It would be helpful to have some key species mentioned

or perfect if you have a vegetation survey of the plots (not mandatory, but any information on that might be very helpful for the user of your data).

**R1.A3:** As mentioned in response to comment R1.C1 we will add information on the vegetation height at the timing of the rainfall experiments in data file "1_site_data.txt". We agree that a vegetation survey and especially the identification of indicator plants could help to classify the individual pasture plots concerning general site-specific soil moisture and wetness conditions. Unfortunately, we cannot provide specific taxonomic information on plants and vegetation of the experimental plots as they were not collected in the field.

**R1.C4:** Please discuss in a broader context which soil parameters (i.e. soil organic matter SOM, soil stone content) or plant parameters might be important and/or influence infiltration behavior as well as water storage capacity. Also address the kinetic energy of your water compared to natural rainfall and how this might impact your results, infiltration, and possible hydrophobic effects - this helps the reader to better understand the complex issue of sprinkler experiments. To cover abovementioned issues I recommend to have a look into papers from Abdallah Alaoui (CH) or Christian Newesely (AUT) or Georg Leitinger (AUT) and many others. Not necessarily cite them but to gain a broader view on the addressed issue.

**R1.A4:** We are aware of the numerous factors that influence runoff generation during extreme rainfall conditions. Therefore we tried to measure a wide range of parameters describing the conditions of the experimental plots. Concerning the kinetic energy of rainfall drops see our reply to R1.A1. Hydrophobic conditions were not observed in any of the 23 experimental sites. We will include a respective comment in the revised manuscript to section "3.4 Field soil description and laboratory analysis". Soil organic matter (Vol-%) and estimated stone content are available for most of the experimental plots. We will add respective data to the file "1_site_data.txt".

**R1.C5:** Figure 1: what do the hatched areas mean? (I assume to highlight which soil types have been investigated / covered, but then: why is there a red dot in a hatched area?). Please clarify!

**R1.A5:** The hatched areas represent common soil types in the federal state of Baden-Württemberg according to the soil map BK 50. This information is missing in the figure description and will be added in the revised manuscript. The red dot represents an experimental site with a certain soil type which was not represented by the soil map BK50 but identified in the field.

**Reply to Review #2:**

Dear reviewer,

Thank you for your valuable comments and recommendations to our manuscript. Please find your questions and comments marked as e.g. "R1.C1" followed by our reply marked as e.g. "R1.A1" including description of changes in the manuscript.

Best regards
Fabian Ries (on behalf of all co-authors)

**R2.C1:** The manuscript presents the experiments in a concise way. It would improve from more details on the test sites and from a more detailed description of the results. The manuscript lacks completely any interpretation and comparisons to similar experiments.

> **R2.A1:** Following also comments of reviewer 1 we will add further data on vegetation height, slope aspect, soil organic matter, stone content and water storage capacity of each individual plot in the data file "1_site_data.txt". Concerning the suggestion of the reviewer to add interpretation and comparison we would like to refer to the journals aims and scope that explicitly state that: "Any interpretation of data is outside the scope of regular articles". Nevertheless, we are currently in the process of publishing results based on this data set in another journal which is currently under review.

**R2.C2:** The authors should mention that similar sprinkling experiments on the hillslope scale were already carried out by others. Please name some of the most important sprinkling experiment studies. How does your experimental set-up and your results compare to the findings of others?

> **R2.A2:** In line with the reviewer's comment we will mention sprinkling experiments from other studies and briefly compare their experimental setup to ours in the introduction section. The comparison of our results to other studies however is outside the scope of this journal (see response R2.A1).

**R2.C3:** It would be useful to have more information on the test sites, e.g. what was the soil depth above the soil-bedrock-interface? Was the bulk density and the density of macropores evaluated?

**R2.A3:** Thank you for this comment. As mentioned in the manuscript we collected data on bulk density, stone content and macropore density and will add respective data to the data file "1_site_data.txt". We did not measure depth to the soil-bedrock interface. Nevertheless, we installed piezometers up to a depth of 90 cm at most locations without reaching the soil-bedrock interface. We will add a respective comment in the revised manuscript.

**R2.C4:** The manuscript would improve from a more detailed description of the results. In particular, a more detailed overview and a comparison of the results on the different test sites would be helpful.

**R2.A4:** A manuscript including a more detailed description of results, comparison and analysis is currently in the process of being published. Our intent of publishing our observations in a data paper is to encourage others to freely make use of the data set for their own research interests. Thus, we would like to refrain from a detailed intercomparing of individual experimental sites and analysis.

**R2.C5:** In addition some interpretation of the results would be very interesting. In the introduction section pluvial floods are mentioned as a motivation for the study. What is the interpretation of your results with regard to pluvial floods? The runoff coefficients show large differences at the different sites. How can this be explained? The runoff coefficients show partially extremely high values of 100% (and more?). How can this be explained?

**R2.A5:** Concerning an interpretation of the results we would like to refer to response R2.A2. For some experiments, runoff coefficients of individual measurement intervals exceeded 100% due to measurement errors in rainfall and runoff rates or spatial rainfall interpolation. However, experimental runoff coefficients were always below 100%. We will add a respective sentence to the manuscript to chapter "4.2 Runoff measurements".

**R2.C6:** Equation 1 and axis labels of Figure 4 are not readable.

**R2.A6:** We will increase font size of Equation 1 and the axis labels of Figure 4 and add axis descriptions to Figure 4.

**R2.C7:** The data set is partially incomplete with regard to soil moisture data, precipitation input and information on site and experiment number.

**R2.A7:** The time series with the individual variables of each location in file "3_experiment_time_series.txt" starts with the day of the first experiment. This structure enables a systematic reading of the respective data sets. The experiment number as well as calculated spatial rainfall values for the subplots are only given for the time period of the respective experiment plus 10 minutes to include the runoff recession into e.g. water balance calculations. Soil

moisture and other variable values are recorded starting with the installation of the respective sensors. Time periods before installation are indicated by NA values. Only few sensors failed at recording values for short periods of time during some of the experiments which are likewise marked with NA. A description of the individual variables of each data file are provided in the file headers. This information is missing in the manuscript. We will add additional information on how to read the data in section "5 Data availability and structure".

**List of changes:**

1. Changed affiliation of co-author Lara Kirn (authors affiliation section)

2. Actualized DOI (abstract section)

5  3. Added paragraph on other studies related to sprinkling experiments following comment R2.C2 (introduction section)

4. Modified description to figure 1 following comment R1.C5 (experimental sites section)

5. Added phrase on depth of soil-bedrock interface following comment R2.C3 (field experiment section)

6. Added phrase on hydrophobic conditions following comment R1.C4 (field experiment section)

7. Added phrase on measurement of organic matter and total porosity following comment R1.C4 (field experiment section)

10  8. Enlarged font size of equation 1 following comment R2.C6 (Data section)

9. Modified figure 4 following comment R2.C6 (Data section)

10. Added explanation on runoff coefficients > 100% following comment R2.C5 (Data section)

11. Actualized DOI (data availability section)

12. Added description on data structure following comment R2.C7

15  13. Added data to file "1_site_data.txt" and uploaded to data repository with a new DOI following requests from both referees.

14. Added various references that were newly included in the manuscript following comments of both referees.

[revised manuscript text omitted]

**Kommentiert [FR8]:** Modified figure with enlarged axis font size and axis descriptions (following comment R2.C6:)

**4.2 Runoff measurements**

An example of the measured variables provided in the data set at two different experimental sites and for all 6 individual experiments (E1-E6) on grassland (site 11) and agricultural field (site 20) are shown in Fig. 5 and Fig. 6. Figure 5 shows the delayed runoff response on the grassland site where overland flow was first observed 30 minutes into the second 100-year
15 rainfall event. The second example (Fig. 6) in contrast, shows overland flow starting only 10 minutes following the initiation of the first sprinkling experiment and quickly reaching a rate close to the rainfall input.

[revised manuscript text omitted]

**Competing interests**

The authors declare that they have no conflict of interests.
* * *
**Kommentiert [FR10]:** Added description of data structure following comment R2.C7

**Kommentiert [FR11]:** Data added following requests from both referees:
- Slope aspect
- Vegetation height
- Stone content
- Macropore density
- Soil organic matter
- Total porosity as a proxy for water storage capacity

**Acknowledgements**

This study was funded by the State Office for the Environment Baden-Württemberg (LUBW). We would like to thank the local water supplier, farmers and landowners for the constructive cooperation and the permission to realize the field experiments on their lands. We owe gratitude to Frank Bartmann from BNNETZE for advice on water abstraction from the
5   public drinking water networks and to Dr. Frank Waldmann from the Federal State Office of Geology, Raw Materials and Mining (LGRB) for discussions on soils and soil-hydrological characteristics in the study area. Especially we would like to thank Marvin Lorff, Haytham Zireeni, Johanna Geilen, Laura Vecera, Annemarie Hoffmann, Lisa Kiemle, Carolin Winter, Jakub Jeřábek, Maja Gensow, Birgit Müller, Emil Blattmann, Jonas Zimmermann and Britta Kattenstroth for greatly supporting the laborious field campaigns.

10   **References**

[revised manuscript text omitted]